# ADMISSIBILITY AND UNIFICATION IN A CERTAIN CLASS OF MULTIMODAL LOGICS

**Nikita A. Protsenko** [*]
School of Mathematics and Computer Science
Siberian Federal University
Krasnoyarsk, 660041, RU
nikitaprotsenko2003@gmail.com

**Vladimir V. Rybakov** [†]
School of Mathematics and Computer Science
Siberian Federal University
Krasnoyarsk, 660041, RU
vrybakov@sfu-kras.ru

## ABSTRACT

This paper investigates unification and admissibility of inference rules in a novel class of multimodal logics designed for multi-agent systems with dynamic information quality. We introduce a logic with cluster-based semantics and moderator agents, where the $Next$ and $Prev$ operators model the flow of time and knowledge revision. Our main technical contribution is the construction of a projective unifier for any unifiable formula in this logic, which allows us to prove that the logic is almost structurally complete - i.e., every admissible inference rule with unifiable premises is derivable. We can also conclude that the admissibility problem in this logic is decidable.

## 1 INTRODUCTION

This paper is devoted to the study of the unification problem for the logic we have constructed. The topic of unification in various logical systems has received significant attention (see, for example, Balbiani & Gougeon (2025); Dzik et al. (2025); Beklemishev (2025)). Although at first glance the problem of unification and the field of non-classical logics may seem far from the tasks of artificial intelligence, an active search for their applications is currently underway. The logic systems being developed, firstly, provide formal frameworks for modeling multi-agent systems, including those capable of disaster recovery (see, for example, Van Ditmarsch & Kuznets (2025)). Secondly, the study of their computational properties, axiomatization, and other abstract characteristics is of direct importance for understanding the limits of expressibility and complexity of reasoning in such systems (see, for example, the works of Jin et al. (2025); Liu & Lorini (2023) and many others).

In this paper, we introduce and elaborate the concept of boundedly reliable and unreliable information. In the context of modal logics, this interpretation can be implemented in various ways (see, for example, Kiyatkin & Rybakov (2024); Protsenko & Rybakov (2025; 2024)). In our system, we introduce a moderator agent that has the ability to view any system state further ahead in the timeline. The system states themselves are structured into clusters. The use of clusters is a common technique in knowledge logic (see, for example, Hintikka (1962); Fagin et al. (2004); Van Ditmarsch et al. (2015)), as they allow one to naturally model the nuances of an agent's awareness. However, unlike the classical approach, we do not consider individual clusters in isolation, but rather analyze their sequences.

The study of the admissibility of inference rules in the constructed model allows us to raise the question of the limits of applicability of knowledge obtained from various clusters and to study the dynamics of changes in the quality of information. Looking at the clusters in sequence, we can see how "unreliable" information can transform into "limited-reliable" information as evidence accumulates from the moderator or during the transition between clusters. Thus, we get a tool for formally describing the processes of learning and revising beliefs: an agent can revise his knowledge of the world if the sequence of his observations becomes unacceptable within the framework of his current theory.

---
[*]https://orcid.org/0009-0001-5435-5785
[†]https://orcid.org/0000-0002-6654-9712

The paper is organized as follows. Section 2 defines the syntax and semantics of our logic. In Section 3, we recall the necessary definitions of unification and admissibility, construct a projective unifier, and prove our main result. Section 4 discusses related works.

## 2 Logic

### 2.1 Syntax

To define formulas of $L$, we fix an enumerable set $Var := \{p_1, p_2, p_3, \dots\}$ of propositional variables. The formulas over the propositional language:

$$\mathcal{L} := \left\langle \bot^0, \top^0, \wedge^2, \vee^2, \neg^1, \square^1, \square_T^1, \Diamond^1, \Diamond_T^1, N^1, P^1 \right\rangle.$$

In Backus-Naur form, the logical system $L$ under consideration is expressed as:

$$\varphi ::= p \mid \bot \mid \top \mid \varphi \wedge \varphi \mid \varphi \vee \varphi \mid \neg\varphi \mid \square\varphi \mid \square_T\varphi \mid \Diamond\varphi \mid \Diamond_T\varphi \mid N\varphi \mid P\varphi$$

The set of all $L$-formulas is denoted by $Fm$. For a formula $\varphi$, $Var(\varphi)$ will denote the set of all variables occurring in $\varphi$.

### 2.2 Frame

*Remark (intuition).* Before giving formal definitions, we explain the intuitive meaning of our framework. We model a multi-agent system evolving through discrete time steps $i = 0, 1, 2, \dots$ At each step, there is a set of possible states of the world (a cluster $C_i$) and a special moderator agent $a_i$ who can observe all future states. The relation R connects all states within a cluster (agents have common knowledge) and connects each state to all future states and moderators (the system remembers the past and can foresee the future). The operator N (Next) moves the system one step forward in time, while P (Prev) moves one step back. The modal operator $\square_T$ captures the knowledge of the moderator: it quantifies only over cluster states, excluding other moderators, reflecting that moderators observe the world, not each other.

We employ relational (Kripke) semantics for the investigation and construction of logical systems. We define a frame for our logic $L$ as the following tuple: $\mathcal{F} := \langle W, R, Next, Prev \rangle$, where

- $W$ is a set of the form $\{a_i \mid i \in \omega\} \cup \bigcup_{i \in \omega} C_i$, ($\omega$ is the natural numbers) where each $C_i$ is some countable or finite (and necessarily non-empty) set of points;
- $R \subseteq W \times W$:
    1. Connection of a node to its own cluster: $\forall i \in \omega \, \forall c \in C_i : (a_i, c) \in R$;
    2. Equivalence within a cluster: $\forall i \in \omega \, \forall c_1, c_2 \in C_i : (c_1, c_2) \in R$;
    3. Connections between clusters and moderators (strictly unidirectional): $\forall i, j \in \omega \, (i < j):$
        (a) $\forall c \in C_i : (c, a_j) \in R$;
        (b) $\forall c \in C_i \, \forall d \in C_j : (c, d) \in R$;
        (c) $(a_i, a_j) \in R$;
        (d) $\forall d \in C_j : (a_i, d) \in R$;
    4. Reflexivity of nodes (moderators): $\forall i \in \omega : (a_i, a_i) \in R$;
    5. $\forall x, y, z \in W : (x, y) \in R \wedge (y, z) \in R \rightarrow (x, z) \in R$;

    That is, $R$ is the smallest relation satisfying conditions 1-4 and closed under transitivity.
- $Next$ is the "next state" relation, formally:
    1. $(a_i, b) \in Next$, for each $b \in C_i$ and for all $i \in \omega$;
    2. $(b, a_{i+1}) \in Next$, for each $b \in C_i$ and for all $i \in \omega$;
- $Prev$ is the "previous state" relation, formally:
    1. $(b, a_i) \in Prev$, for each $b \in C_i$ and for all $i \in \omega$;
    2. $(a_{i+1}, b) \in Prev$, for each $b \in C_i$ and for all $i \in \omega$;
    3. $(a_0, a_0) \in Prev$;

## 2.3 GRAPHICAL INTERPRETATION OF FRAMES

We provide a graphical interpretation of a fragment of a possible frame. In this case, we have depicted two two-element clusters, $C_0$ and $C_1$, and two points $a_0, a_1$, which correspond to these clusters. The relation $Next$ is also represented here.

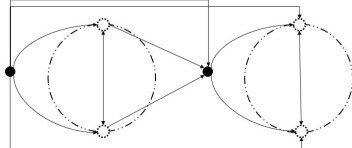

Figure 1: Fragment of a frame with two clusters $C_0 = \{c_0, c_1\}$ and $C_1 = \{d_0, d_1\}$ (the cluster points are indicated by a dotted line with a white background) and moderator nodes $a_0$, $a_1$ (they are indicated by black dots in the drawing). Solid arrows represent the R relation. Additionally, the points inside the cluster are located on a large dotted circle.

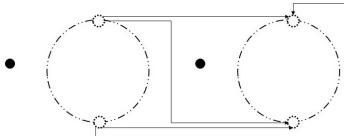

Figure 2: The relation $R$ (between clusters)

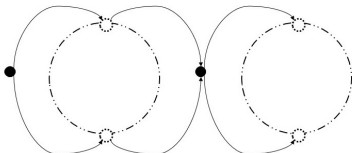

Figure 3: The relation $Next$

## 2.4 MODELS. SATISFIABILITY.

A model for the logic $L$ is defined as a tuple $M := \langle \mathcal{F}, V \rangle$, where $V : Prop \mapsto \mathcal{P}(W)$ is a valuation of propositional variables. In this paper, we restrict our attention to a single valuation, although logics with multiple valuations (multi-valuation) - which can also capture a variety of interesting scenarios - have been studied in the literature.

We define the satisfiability relation on the states of a model inductively. We write $(M, x) \models_V \varphi$ to denote that a formula $\varphi$ is true in the model $M$ at state $x$ under the valuation $V$:

1. $(M, x) \models_V p \qquad \Leftrightarrow x \in V(p)$;
2. $(M, x) \models_V \top$;
3. $(M, x) \models_V \varphi \wedge \psi \qquad \Leftrightarrow (M, x) \models_V \varphi \text{ AND } (M, x) \models_V \psi$;
4. $(M, x) \models_V \varphi \vee \psi \qquad \Leftrightarrow (M, x) \models_V \varphi \text{ OR } (M, x) \models_V \psi$;
5. $(M, x) \models_V \neg\varphi \qquad \Leftrightarrow (M, x) \not\models_V \varphi$;
6. $(M, x) \not\models_V \bot$;
7. $(M, x) \models_V \Box_T \varphi \qquad \Leftrightarrow (\forall y \in W \setminus \{a_0, a_1, \dots\})((x, y) \in R \to (M, y) \models_V \varphi)$;
8. $(M, x) \models_V \Box\varphi \qquad \Leftrightarrow (\forall y \in W)((x, y) \in R \to (M, y) \models_V \varphi)$;
9. $(M, x) \models_V N\varphi \qquad \Leftrightarrow (\exists y)((x, y) \in Next \wedge (M, y) \models_V \varphi)$;

10)  $(M, x) \models_V P\varphi \qquad \Leftrightarrow (\exists y)((x, y) \in Prev \wedge (M, y) \models_V \varphi);$

*Explanation 7)-8):* The operator $\square_T$ quantifies exclusively over cluster points $(C_i)$, excluding moderator nodes from consideration. However, due to the frame conditions, a moderator $a_i$ has $R$-access not only to its own cluster $C_i$, but also to all future clusters $C_j$ $(j > i)$. Therefore, when evaluated at a moderator $a_i$, the formula $\square_T\varphi$ means that $\varphi$ holds at *every point in every cluster reachable from $a_i$* i.e., in $C_i, C_{i+1}, C_{i+2}, \ldots$ This captures the idea that the moderator has a "forward-looking" perspective: it can inspect its current cluster and receive information from all subsequent clusters, but it cannot access the knowledge of other moderators directly, nor can it access past clusters.

When evaluated at a cluster point $c \in C_i$, the situation is different: from $c$, the $R$-relation provides access to all points in $C_i$ (due to intra-cluster equivalence) and to all points in future clusters $C_j$ $(j > i)$ and future moderators $a_j$ (due to condition 3). The $\square_T$ operator at $c$ thus quantifies over all cluster points reachable from $c$, which includes $C_i$ and all future $C_j$, but excludes the moderators $a_j$ themselves.

*Remark (intuition).* The relation $Next$ models a complete cycle of the system's operation: first, the moderator $a_i$ inspects (checks) the current state of its cluster $C_i$ (the transition $(a_i, c)$), and then each of these states initiates a transition to the next moderator $a_{i+1}$ (the transition $(c, a_{i+1})$). The relation $Prev$ is defined as the converse of $Next$, with the addition of a self-loop $(a_0, a_0)$ to ensure that the operator $P$ is meaningful at the initial point. The absence of a transition $(c, a_0)$ for $c \in C_0$ underscores that $a_0$ represents the absolute beginning of time.

For $\lozenge$ and $\lozenge_T$, we introduce the following abbreviations: $\lozenge_T := \neg\square_T\neg$ and $\lozenge := \neg\square\neg$.

The logic $L$ is then defined semantically as follows:

$$L := \{\varphi \mid \forall M \, \forall x \in W : (M, x) \models_V \varphi\}.$$

## 2.5 Examples and Possible Interpretations

**Example 1.** Let the proposition $p$ mean "the agent votes in favor".

Formula: $\square_T(p \rightarrow \lozenge_T\neg p) \wedge \lozenge_T(p \wedge \square_T\lozenge p) \wedge \square_T\lozenge_T p$

This formalizes how agents in a MAS reach consensus in the presence of guaranteed opposition. Any agent voting "in favor" must be aware that there is at least one vote "against" in the system. There exists an agent who votes "in favor" and is confident that in all future rounds there will be voters "in favor". Every agent observes that in the future there will be voters "in favor".

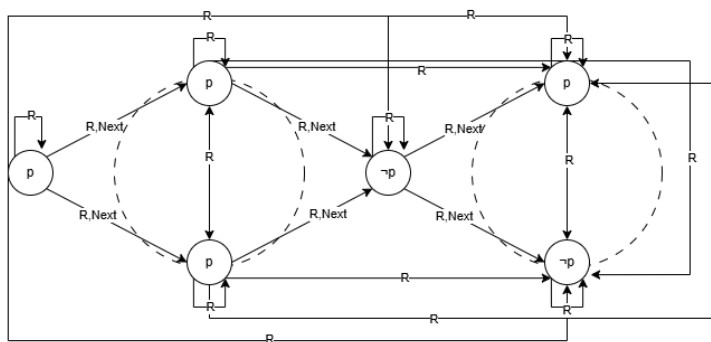

Figure 4: Example model $M$, where $(M, a_0) \models_V \square_T(p \rightarrow \lozenge_T\neg p) \wedge \lozenge_T(p \wedge \square_T\lozenge p) \wedge \square_T\lozenge_T p$

$W = \{a_0, c_0, c_1, a_1, d_0, d_1\}$, $\quad C_0 = \{c_0, c_1\}$, $\quad C_1 = \{d_0, d_1\}$, $Next = \{(a_0, c_0), (a_0, c_1), (c_0, a_1), (c_1, a_1), (a_1, d_0), (a_1, d_1)\}$, $\quad R = Next \cup \{(a_0, a_1), (a_0, d_0), (a_0, d_1), (c_0, c_1), (c_0, d_0), (c_0, d_1), (c_1, c_0), (c_1, d_0), (c_1, d_1)\} \quad \cup \quad Ref$, $V(p) = \{a_0, c_0, c_1, d_0\}$

**Example 2.** Let the proposition $p$ mean "the model recognizes the pattern".

Formula: $\Box_T(p \to \Box_T p) \land \Diamond_T p \land \Diamond_T \neg p$

Knowledge, once acquired, is retained forever.

**Example 3.** Let the proposition $p$ mean "the agent has chosen strategy A".

Formula: $\Box_T(p \to Np) \land \Box_T(\neg p \to N\neg p) \land \Diamond_T p \land \Diamond_T \neg p$

Truth at each level determines truth at the next level of control.

## 3 Unification. Admissibility.

### 3.1 Key Theorems and Definitions

A inference rule r is an expression: $r := \frac{\varphi_1(p_1,\ldots,p_n),\ldots,\varphi_m(p_1,\ldots,p_n)}{\psi(p_1,\ldots,p_n)}$. The inference rule $r$ is said to be valid in the Kripke structure $\mathcal{F}$ (notation $\mathcal{F} \models r$) if $(\mathcal{F} \models \land_{1 \le i \le m} \varphi_i) \Rightarrow \mathcal{F} \models \psi$.

Inference rule $r$ is said to be admissible in $L$ if, $\forall \alpha_1 \in Fm, \ldots \forall \alpha_n \in Fm$ $\land_{1 \le i \le m}[\varphi_i(\alpha_1,\ldots,\alpha_n) \in L] \Rightarrow [\psi(\alpha_1,\ldots,\alpha_n) \in L]$

We say that a formula $\varphi$ is *unifiable* in a logic $L$ if there exists a substitution $\sigma$ such that $\sigma(\varphi)$ is a theorem of $L$, i.e., $\sigma(\varphi) \in L$.

A particular type of unifier is a *ground unifier*, which is a substitution that assigns to each variable either $\top$ (true) or $\bot$ (false). It is usually denoted by $gu$.

We say that $\sigma$ is a *projective unifier* if $\sigma$ is a unifier and

$$\Box\varphi \to [p_i \leftrightarrow \sigma(p_i)] \in L,$$

for all $p_i \in Var(\varphi)$.

The logical connective $\leftrightarrow$ is defined as

$$\varphi_1 \leftrightarrow \varphi_2 := (\varphi_1 \to \varphi_2) \land (\varphi_2 \to \varphi_1).$$

**Theorem (see Dzik (2011), Corollary 6).** If a logic has projective unifiers, then it is almost structurally complete. In other words, every admissible inference rule with unifiable premises is derivable in this logic.

### 3.2 Constructing a Projective Unifier

Consider a unifiable formula $\varphi$ and expressions of the form

$$\psi_\varphi(X) := \bigwedge_{p_i \in X \cap Var(\varphi)} p_i \land \bigwedge_{p_i \in Var(\varphi) \setminus X} \neg p_i.$$

Consider all possible subsets $X \subseteq Var(\varphi)$ and enumerate all such formulas as $\alpha_1, \alpha_2, \ldots, \alpha_{|\mathcal{P}(Var(\varphi))|}$.

We consider two possible cases for an arbitrary point $a$ in a model $M$. In the first case, the formula $\Diamond\Box\varphi$ holds at every point $a$ of the model $M$. In the second case, the formula $\neg\Diamond\Box\varphi$ holds at every point $a$ of the model $M$.

**Lemma.** If at some state $a$ of a model $M$ we have $(M,a) \models_V \Diamond\Box\varphi \land \neg\Box\varphi$, then there exists a point $b$ such that $(M,b) \models_V \Box\varphi \land P\neg\varphi$ and $(a,b) \in R$.

Moreover, we can pinpoint exactly which formula among $\alpha_1, \alpha_2, \ldots, \alpha_{|\mathcal{P}(Var(\varphi))|}$ must be true at this point $b$. It is precisely on the basis of this formula $\alpha_j$ that we will construct our substitution.

To achieve this, we introduce a priority selector formula $\Theta_j$ based on lexicographic ordering. It is true at a point if and only if $\alpha_j$ is the available witness valuation with the smallest index:

$$\Theta_j := \Diamond(\Box\varphi \land P\neg\varphi \land \alpha_j) \land \bigwedge_{m=1}^{j-1} \neg\Diamond(\Box\varphi \land P\neg\varphi \land \alpha_m)$$

*(Note: for $j = 1$, the empty conjunction $\bigwedge_{m=1}^{0}$ is treated as $\top$, so $\Theta_1 = \Diamond(\Box\varphi \wedge P\neg\varphi \wedge \alpha_1)$).*

Now, for every unifiable formula $\varphi(p_1, \ldots, p_n)$, we define the substitution $\sigma$ as follows:

$$\sigma(p_i) := \left[\Box\varphi \wedge p_i\right] \vee \left[\neg\Diamond\Box\varphi \wedge gu(p_i)\right] \vee \left[\neg\Box\varphi \wedge \Diamond\Box\varphi \wedge \bigvee_{j \in J(p_i)} \Theta_j\right],$$

where $gu$ is a ground unifier for $\varphi$, and $J(p_i)$ is the set of indices of those conjunctions in which $p_i$ occurs positively (without negation):

$$J(p_i) := \{j \in \{1, \ldots, N\} \mid p_i \text{ is true in } \alpha_j\}$$

**Example.** Let $\varphi := \Box\neg p$. Since $\varphi$ is unifiable, we fix the ground unifier $gu(p) := \bot$. The set of variables is $Var(\varphi) = \{p\}$. We define the strict enumeration of all possible truth assignments (maximal conjunctions) as $\alpha_1 := p$ and $\alpha_2 := \neg p$.

Consider a model $M$ with points $a, c, b$ such that:

1. $(M, a) \models \neg\Box\varphi \wedge \Diamond\Box\varphi$ (Point $a$ is in the transition zone);
2. $R$ relations: $aRc$ and $cRb$ (so $aRb$ by transitivity);
3. $(M, c) \models p$ and $(M, c) \not\models \Box\varphi$ (Transition point);
4. $(M, b) \models \Box\varphi \wedge \neg p$ (Stable witness point).

To verify $(M, a) \models \sigma(\varphi)$, we check $(M, a) \models \Box\neg\sigma(p)$. This requires evaluating $\sigma(p)$ at all reachable points $y \in \{c, b\}$.

Since $p$ occurs positively only in $\alpha_1$, the set of target indices is $J(p) = \{1\}$. The priority selector for $\alpha_1$ is $\Theta_1 = \Diamond(\Box\varphi \wedge P\neg\varphi \wedge p)$. Thus, the substitution is defined as:

$$\sigma(p) = \underbrace{\left[\Box\varphi \wedge p\right]}_{D_1} \vee \underbrace{\left[\neg\Diamond\Box\varphi \wedge gu(p)\right]}_{D_2} \vee \underbrace{\left[\neg\Box\varphi \wedge \Diamond\Box\varphi \wedge \Theta_1\right]}_{D_3}$$

We analyze the active disjunct at each reachable point:

**At point $c$ (Transition Zone):** Here $\neg\Box\varphi$ and $\Diamond\Box\varphi$ hold. Thus, $D_1$ and $D_2$ are false. The active term is $D_3$, which depends entirely on $\Theta_1$. We expand it as follows:

$$D_3(c) \iff \top \wedge \top \wedge \Diamond\left[\Box\varphi \wedge P\neg\varphi \wedge p\right]$$

Let us evaluate $\Theta_1$ at $c$:
- The operator $\Diamond$ looks for a reachable witness point satisfying $W \wedge p$.
- From $c$, the only reachable point satisfying the witness condition $W$ is $b$.
- However, at our witness $b$, the variable $p$ is false ($\neg p$ holds). Therefore, $(M, b) \not\models W \wedge p$.
- Since no reachable witness satisfies $p$, the selector $\Theta_1$ evaluates to $\bot$.

*(Note: The selector $\Theta_2$ for $\alpha_2$ would be true here, but it is not included in the disjunction for $\sigma(p)$ because $p$ is negative in $\alpha_2$. This strictly forces $\sigma(p)$ to adopt the false valuation).* Thus, $(M, c) \models \sigma(p) \leftrightarrow \bot$.

**At point $b$ (Stable Zone):** Here $\Box\varphi$ holds. Thus, $D_2$ and $D_3$ are false. The active term is $D_1$:

$$(M, b) \models D_1 \iff (M, b) \models \Box\varphi \wedge p$$

Since $(M, b) \models \neg p$, this evaluates to $\bot$.

$$(M, b) \models \sigma(p) \leftrightarrow \bot$$

**Conclusion:** Since $\sigma(p)$ evaluates to $\bot$ at all points reachable from $a$:

$$(M, a) \models \Box\neg(\sigma(p)) \iff (M, a) \models \Box\neg(\bot) \iff (M, a) \models \Box\top$$

Therefore, $(M, a) \models \sigma(\varphi)$.

### 3.3 Main Result

**Theorem.** $\sigma$ is a projective unifier for every unifiable formula $\varphi$.

*Idea:*

The proof proceeds by case analysis on an arbitrary point $a$ in an arbitrary model $M$.

**Case 1: $\Box\varphi$ holds at $a$.** Then $\varphi$ itself holds at $a$. The substitution $\sigma$ reduces to its first disjunct, so the valuation of variables remains unchanged. Hence $\sigma(\varphi)$ holds at $a$.

**Case 2: $\Box\varphi$ fails at $a$.** This splits into two subcases:

> **Subcase 2.1: $\neg\Diamond\Box\varphi$ holds at $a$.** Only the second disjunct of $\sigma$ (the ground unifier part) is relevant. Since $\varphi$ is unifiable, there exists a ground unifier $gu$ that makes $\varphi$ a theorem. Thus $\sigma(\varphi)$ holds at $a$.

> **Subcase 2.2: $\Diamond\Box\varphi$ holds at $a$.** The third disjunct of $\sigma$ is active. By the Lemma, there is at least one witness point $b$ reachable from $a$ such that $\Box\varphi \wedge P\neg\varphi$ is satisfied.
>
> Because points in a cluster are $R$-equivalent, there might be multiple such witness points in the future cluster, potentially satisfying different conjunctions $\alpha$. If variables evaluated $\Diamond$ independently, $\sigma$ could produce a mixed valuation at $a$ that did not exist in any single witness, falsifying $\varphi$.
>
> The priority selector $\Theta_j$ rigorously prevents this. Evaluated at point $a$, exactly one formula $\Theta_k$ among $\{\Theta_1, \ldots, \Theta_N\}$ will evaluate to $\top$ - specifically, the one corresponding to the available witness valuation $\alpha_k$ with the lowest index $k$. All other $\Theta_m$ ($m \neq k$) will rigorously evaluate to $\bot$ at $a$.
>
> Consequently, the substitution acts as a strict filter. For any variable $p_i$:
>
> - If $p_i$ is positive in the universally selected $\alpha_k$, then $k \in J(p_i)$. The term $\Theta_k$ is present in the disjunction for $\sigma(p_i)$, making $\sigma(p_i) = \top$ at $a$.
> - If $p_i$ is negative in $\alpha_k$, then $k \notin J(p_i)$. The true term $\Theta_k$ is absent from the disjunction, and since all other $\Theta$ are $\bot$, $\sigma(p_i)$ evaluates to $\bot$ at $a$.
>
> Effectively, this mechanism securely identifies a single valid valuation $\alpha_k$ from the witness cluster and replicates it perfectly at the original point $a$, without any distortion. This ensures that $\sigma(\varphi)$ seamlessly evaluates to $\top$ at $a$.

In all cases, $\sigma(\varphi)$ is true at every point of every model, making it a theorem. The projective property follows from the fact that $\sigma$ either preserves the original valuation (Case 1) or aligns it with a ground unifier or a designated witness point (Cases 2.1 and 2.2), satisfying the condition $\Box\varphi \rightarrow (p_i \leftrightarrow \sigma(p_i))$.

**Theorem.** Admissibility problem in this logic is decidable

**Theorem.** The logic $L$ is almost structurally complete.

## 4 Related works

As discussed earlier in this work, we have only considered cluster structures that are arranged linearly. This concept can undoubtedly be extended to a broader class, for example, to interval structures. That is, to restrict the interaction between clusters to the boundaries of intervals (see, for example Protsenko et al. (2023)). Equally interesting is the additional investigation of decidability problems from a computational perspective, i.e., determining the complexity class. We also foresee further development in creating a framework that would model the operation of some multi-agent system based on the proposed logical systems.

A second, equally important component consists of extending the studied logical system with operators that can explicitly model uncertainty, possibly through frames of a special kind or through models with multiple valuations. Indeed, if we consider, for example, operators that count some parameter, then the question that remains open is not only that of decidability, but also more complex ones, such as structural completeness.

ACKNOWLEDGMENTS

This work was supported by the Russian Science Foundation and Krasnoyarsk Regional Fund of Science (Project No 25-21-20011, https://rscf.ru/en/project/25-21-20011/).

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
