# OpenReview forum: "Admissibility and Unification in a Certain Class of Multimodal Logics"
_mathai.club/MathAI/2026/Conference — 2026 Oral_

### Official Review · Reviewer_7q7g · 2026-03-13
**Good result, but the presentation needs to be improved.**

**Rating:** 9
**Confidence:** 4

**Review:**

This paper introduces a new multimodal logic designed to model multi-agent systems with a focus on dynamic information quality, moderator agents, and a cluster-based semantics that incorporates temporal operators (Next and Prev). The main technical claim is the construction of a projective unifier for any unifiable formula, which the authors use to argue that the logic is structurally complete (i.e., all admissible rules are derivable). While the paper addresses an interesting and relevant topic at the intersection of modal logic, unification theory, and multi-agent systems, its current form requires some revision. The main drawback is the lack of an intuitive example explaining the concepts of the proposed logic.
Minor comments:
In line 84, what is ω formally?
Starting from Section 3, there is no numbering of lemmas, examples and theorems.

---

### Official Review · Reviewer_Jxft · 2026-03-13
**Admissibility and Unification in a Certain Class of Multimodal Logics (A Review)**

**Rating:** 7
**Confidence:** 4

**Review:**

The paper investigates unification and admissibility of inference rules in a novel class of multimodal logics designed for multi-agent systems with dynamic information quality. The authors introduce a logic with cluster-based semantics and moderator agents, where the Next and P rev operators model the flow of time and knowledge revision. The main technical contribution is the construction of a projective unifier for any unifiable formula in this logic, which allows to prove that the logic is structurally complete - i.e., every admissible inference rule is derivable. The paper provides a tool for formally describing the processes of learning and revising beliefs: an agent can revise his knowledge of the world if the sequence of his observations becomes unacceptable within the framework of his current theory.

---

### Decision · Program_Chairs · 2026-03-14

**Decision:**

Accept (Oral)

**Comment:**

Dear Author(s),

On behalf of the Program Committee of the International Conference on Mathematics of Artificial Intelligence (MathAI 2026), we are pleased to inform you that your paper has been accepted for an oral presentation at MathAI 2026.

Your paper was evaluated through a rigorous two-stage review process involving both automated screening and expert review by members of the Program Committee. The reviewers recognized the quality and contribution of your work.

Presentation details:

- Format: Oral presentation (15–20 minutes + 5 minutes Q&A)
- Mode: You may present either in person (offline) at the conference venue in Sirius, Russia, or remotely via Zoom. Please indicate your preferred mode when confirming your participation.
- Conference dates: Marh 30 - April 3, 2026
- Website: https://mathai.club

Next steps:

1. Please confirm your participation and presentation mode by replying to this email mathai.club@yandex.ru no later than March 15, 2026 18:00 Moscow time.
2. If you plan to attend in person, the organizing committee will provide accommodation details separately.
3. Please prepare your final camera-ready manuscript according to the formatting guidelines available at https://mathai.club and upload it to OpenReview by March 15, 2026 18:00 Moscow time.

Should you have any questions regarding the program, logistics, or your presentation slot, please do not hesitate to contact us.

We look forward to your contribution to MathAI 2026.

With kind regards,

MathAI 2026 Program Committee
International Conference on Mathematics of Artificial Intelligence
https://mathai.club
OpenReview: https://openreview.net/group?id=mathai.club/MathAI/2026/Conference
Telegram: https://t.me/MathAI_club
Email: mathai.club@yandex.ru